# A Study of the Relationship between Men Who Have Sex with Men Stigma and Depression: A Moderated Mediation Model

**DOI:** 10.3390/healthcare11212849

**Published:** 2023-10-29

**Authors:** Tianyi Zhou, Qiao Chen, Xiaoni Zhong

**Affiliations:** School of Public Health, Chongqing Medical University, Chongqing 400016, China; 2021110620@stu.cqmu.edu.cn (T.Z.); 2021120743@stu.cqmu.edu.cn (Q.C.)

**Keywords:** men who have sex with men, social support, moderated mediation models, depression, enacted stigma, perceived stigma

## Abstract

(1) Background: Men who have sex with men (MSM) have a high prevalence of depression due to stigma. However, whether resilience and social support play a moderating role in the effects of stigma on depression remains to be tested. This study constructed a moderated mediation model to explore the mediating role of perceived stigma in the relationship between enacted stigma and depression and whether this relationship is moderated by social support. (2) Methods: MSM were recruited during November–December 2022 using a non-probability sampling method, and a total of 1091 participants were included. Enacted stigma, perceived stigma, resilience, social support, and depressive symptoms were measured. Mediation and moderated mediation models were used to analyze the relationships between these variables. (3) Results: Moderated mediation analyses show that enacted stigma indirectly affects depression through perceived stigma (β = 0.315, 95% confidence interval = 0.221 to 0.421). Social support had a positive moderating effect between enacted stigma and depressive symptoms (β = 0.194, *p* < 0.001) and a negative moderating effect between perceived stigma and depressive symptoms (β = −0.188, *p* < 0.001). (4) Resilience and perceived stigma mediated the relationship between enacted stigma and depression, and the relationship between enacted stigma, perceived stigma, and depression was moderated by social support. Reducing stigma while increasing social support has the potential to alleviate depressive symptoms among Chinese MSM.

## 1. Introduction

Globally, men who have sex with men (MSM) have significantly poorer mental health compared to the general population, with a particularly high prevalence of depressive symptoms. Results from systematic regression and meta-analysis showed that the overall prevalence of depression among MSM globally was 35% (95% CI = 31% to 39%, *p* = 0.001) [1]; in a sample of 21,950 Chinese men who have sex with men, the overall prevalence of depression was 40% (95% CI = 37.9% to 45.0%) [2]. Overall, the prevalence of depressive symptoms is two to three times higher among men who have sex with men than among heterosexual men [3,4,5]. The prevalence of depression varies somewhat between countries due to a variety of factors. The impact of depressive symptoms on MSM is enormous. As a population at high risk for HIV, depression becomes a major barrier for MSM when seeking health services as well as HIV care [6]. In addition, chronic depression increases the risk of poor lifestyle [7], substance abuse [8], suicide [9], and high-risk sexual behavior among MSM [10].

MSM has a long history in China. Over time, Chinese society’s attitudes towards MSM have changed. Generally speaking, there is no persecution of MSM in China, and there are no laws or policies prohibiting same-sex sexual behavior. In recent years, as the trend of HIV prevalence among MSM in China has risen significantly, this group has received wider attention, and public awareness of MSM has increased. Traditional Chinese culture promotes heterosexual marriage and same-sex marriage and other same-sex partnerships are not legally supported in China, and there are no relevant anti-discrimination policies [11,12]. In 2001, The China Psychiatrics Classification and Diagnostic Criteria excluded homosexuality from mental illness, but the view that same-sex sexuality is a pathology persists. These social characteristics and cultural contexts create a stigmatizing environment that is detrimental to the mental health of Chinese MSM. Unlike in the West, traditional Chinese families emphasize filial piety and procreation [13], which leads them to face more minority stress due to negative attitudes from the family. In addition, Chinese society emphasizes a collectivist culture that respects social norms and order. MSM are relatively invisible in China, and they largely hide their sexual identity in order to gain acceptance. Various factors have led to a higher prevalence of psychological problems among MSM in China than in other countries. Although the personal rights of MSM are receiving increasing attention in China and public attitudes towards this group are becoming more tolerant, the problem of depression among MSM needs to be improved [14].

Depression occurs as a result of a synergistic combination of social, psychological, and biological factors, with sexual identity-related stigma being the most important predictor and determinant [15]. Previous research has shown that MSM stigma is a stronger predictor of depression than stress in general: perceived stigma and enacted stigma were highly prevalent among 736 participants, and stigma was associated with a higher prevalence of depressive symptoms [16]. The results of a cross-sectional study of Chinese MSM show that discrimination and stigma play an important role in the development of depression [17]. The results of another study in Nanjing, China, showed that perceived stigma not only directly affects depressive symptoms but also indirectly affects depression through other factors [18]. In addition, age, internalized homophobia, coping skills, and social support have been shown to be potential determinants of depression among MSM [19,20].

The sexual minority stress model is often used in studies of the relationship between stigma and depression as a targeted classical theory that aims to reveal how MSM’s experiences of sexual stress affect their mental health, helping us to explore the mechanisms of stigma’s effect on depression [21,22]. Meyer describes the particular stressors experienced by MSM as a continuum: distal (discrimination and incidents of violence) to proximal (perceived stigma and internalized homophobia), with these sexual minority stressors ultimately leading to psychiatric disorders, including depression [23].

In addition, with regard to coping with depression, the sexual minority stress model suggests two positive moderators, resilience and social support. Resilience refers to the ability of MSM to adapt in the face of discrimination and stress, and helps to alleviate negative psychological states. Individuals with greater psychological resilience can cope with problems and maintain a healthy psychological state by strengthening their emotional control [24]. Previous research has shown that resilience has a positive moderating effect between the implementation of stigma and depressive symptoms [25]. There is also a study that points out that stigma can indirectly affect mental health through resilience [26].

Social support is another protective factor for depression, broadly defined to include interpersonal relationships in the community, family support, and cultural connections. Creating connections with peers facilitates MSM to share experiences with each other and normalize emotions after experiencing stigma [27]. In a study of young black MSM, participants were more likely to accept help from peers than from mental health professionals, and peer support reduced the risk of adverse conditions for MSM [28]. Lack of support from family members can lead to depression [29,30]. A study in Canada showed that higher support was positively associated with health behaviors and that social support buffered the effects of stigma on psychological conditions [21]. As a valuable social resource, social support has been shown to not only buffer men who have sex with men from depressive symptoms but also to promote physical health status and improve their quality of life [31,32].

In summary, we propose a moderating model based on the sexual minority stress model, as in Figure 1. However, in previous studies, we found that the moderating effect of resilience between stigma and depression was not significant, and similar results were found in a cross-sectional study in India, where resilience mediated the effect of stigma on depression, but did not have a moderating effect [33]. We then considered psychological resilience as a mediator [18], constructed a moderated mediation model, and constructed several hypotheses: (1) perceived stigma and resilience mediated the effects of enacted stigma on depression; and (2) social support moderated the effects of enacted stigma and perceived stigma on depression. The sexual minority stress model has received much validation in Western samples, but there is less research on depression and sexual minority stress among Chinese MSM. This study aims to suggest mental health interventions for MSM in China, and, to some extent, provide insights for improving the mental health of MSM in other parts of the world.

## 2. Materials and Methods

### 2.1. Participants

Between November and December 2022, MSM were recruited using a non-probability sampling method with the following conditions: (1) 18 years of age or older; (2) self-reported that they had engaged in anal or oral sex with a man in the last year; and (3) no serious mental illness or intellectual disability. Staff distributed structured questionnaires to MSM via WeChat, and participants completed the questionnaires anonymously and were rewarded with 15 RMB (approximately US$2.18) each after passing the audit. A total of 1699 participants were recruited for this study, and a total of 608 (35.78%) were excluded due to (1) being under the age of 18 years (18); (2) completing the questionnaire in less than 2 min (433); or (3) failing the logic check (157). A total of 1091 MSM were finally included in the study.

The study was conducted after obtaining informed consent from the participants. Participants had the right to withdraw from the survey at any time, and all participant responses would be kept confidential. This study was approved by the Ethics Committee of Chongqing Medical University (2019001, Chongqing, China).

### 2.2. Predictor Variable

Enacted stigma. The multidimensional MSM identity stigma scale has been used as a stigma measurement tool [34]. The scale has four subscales and contains 18 items. The subscale “Enacted Stigma” was used for measurement, with items such as “You conflict with others because you are an MSM”, “You have lost your job because you are an MSM”, etc. The scale was scored on a 5-point Likert scale, with higher scores indicating more frequent experiences. The subscale Cronbach’s coefficient was 0.895.

### 2.3. Mediator/Moderator Variables

Perceived stigma. The “Anticipated” subscale of the multidimensional MSM identity stigma scale was used [34], with items such as “Family members have some negative attitudes toward you”, “Some friends avoid you”, etc. The Cronbach’s coefficient for the scale was 0.901.

Resilience. The short version of the psychological resilience scale (10-item Connor–Davidson resilience scale, CD-RISC-10) was used to measure resilience [35]. The scale consists of 18 items assessing resilience in three dimensions: resilience, strength, and optimism. Items include “Able to adapt to change” and “Can deal with whatever comes”. The scale is scored on a 5-point Likert scale ranging from 1 (never) to 5 (always), and the total resilience score is the sum of the scores for each item. The Cronbach’s coefficient for the scale was 0.939.

Social support. The multidimensional perceived social support scale was used to measure perceived support from three dimensions: family, friends, and others [36]. The scale consists of 12 items, with items such as “My family tries to help me”, “I can talk to my friends about my problems”, “There is someone who will really give me comfort”, etc. The scale is based on a 5-point Likert scale. The scale is scored on a 5-point Likert scale ranging from 1 (strongly disagree) to 5 (strongly agree), and the total social support score is the sum of the scores for each entry. The Cronbach’s coefficient for the scale was 0.948.

### 2.4. Outcome Variable

Depression. Depression was measured using the Center for Epidemiological Survey depression scale (CES-D) [37]. The scale contains a total of 20 entries. The scale is scored on a 4-point scale from 0 (less than 1 day or never) to 3 (5–7 days). The total score is 60, and depressive symptoms may be present with a score of 20 or more. Items include “I am bothered by little things”, “I don’t want to eat”, and “I have a bad appetite”. The Cronbach’s coefficient for the scale was 0.936.

### 2.5. Control Variables

Control variables include age, education (1 = illiteracy and semi-illiteracy, 2 = grade school, 3 = junior high school, 4 = senior high school, 5 = junior college, 6 = university degree or above), and monthly disposable income (1 = 1000 Ren Min Bi or less, 2 = 1001–3000 RMB, 3 = 3001–5000 RMB, 4 = 5001–10,000 RMB, and 5 = 10,000 RMB or more).

### 2.6. Statistical Analysis

In this study, data were analyzed using SPSS 26.0 for descriptive statistics, SPSS 26.0 PROCESS macro for mediation and mediation with moderation, and Mplus 8.3 to output simple slope plots of moderated effects. First, frequencies and percentages were used to describe sociodemographic characteristics, and correlation analyses between variables were performed. Second, a validated factor analysis of the variables was conducted to test the reliability and validity of the measures. Finally, mediation models (SPSS 26.0 PROCESS macro, MODEL6) were used to test the mediating role of perceived stigma and resilience between enacted stigma and depressive symptoms, and moderated mediation models (SPSS 26.0 PROCESS macro, MODEL89) were used to test the moderating role of social support between enacted stigma, perceived stigma, and depressive symptoms. The bootstrap method (5000 bootstrap samples) was used to test 95% confidence intervals. Age, education, and income were used as control variables in mediation analyses, and mediation analyses with moderation were significant at *p* < 0.05.

## 3. Results

### 3.1. Sample Characteristic

In the sample of 1091 MSM, the mean age was 26.34 (standard deviation = 6.324). 69.4% of the participants were urban residents, and 49.2% had a bachelor’s degree or higher. The majority of MSM were unmarried (69.8%), and more than half were employed (66.7%); 31.5% of MSM had monthly disposable incomes of between RMB 3001–5000, and 31.7% had monthly disposable incomes of between RMB 5001–10,000 (Table 1).

### 3.2. Correlation Analysis

The results showed that perceived stigma was significantly and positively correlated with enacted stigma (R = 0.063, *p* < 0.001), depression was significantly and positively correlated with enacted stigma (R = 0.621, *p* < 0.001), and significantly and positively correlated with perceived stigma (R = 0.475, *p* < 0.001). Age, education, and income were included as control variables (Table 2).

### 3.3. Confirmatory Factor Analysis

A series of validated factor analyses were used to test the reliability and validity of the measurements. The fit indices of the 5-factor validated factor analysis model: X^2^/df = 3.567, RMSEA = 0.049, CFI = 0.928, TLI = 0.924, and SRMR = 0.043 (Table 3). This model is significantly better than the other models, indicating that there is some differentiation between the five variables in this study and that it is possible to proceed to the next step in the study.

The results of Model 3 showed that enacted stigma significantly positively affected depressive symptoms (β = 0.503, *p* < 0.001), perceived stigma significantly positively affected depressive symptoms (β = 0.204, *p* < 0.001), and resilience significantly negatively affected depressive symptoms (β = −0.215, *p* < 0.001). The results of Model 4 showed a total effect of enacted stigma on depressive symptoms (β = 0.604, *p* < 0.001). The total indirect effect of the mediation model was (β = 0.243, 95% CI = 0.136 to 0.352), perceived stigma partially mediated the relationship between enacted stigma and depression (β = 0.315, 95% CI = 0.221 to 0.421), resilience and perceived stigmatization partially mediated the relationship between enacted stigma and depression (β = −0.082, 95% CI = −0.118 to −0.051) (Table 4).

### 3.4. Moderated Mediation Effect Analysis

After controlling for age, literacy, and income, enacted stigma significantly and positively affected depressive symptoms (β = 0.501, *p* < 0.001), perceived stigma significantly and positively affected depressive symptoms (β = 0.170, *p* < 0.001), and resilience significantly and negatively affected depressive symptoms (β = −0.148, *p* < 0.001). In addition, the interaction between enacted stigma and social support had a significant effect on depressive symptoms (β = 0.194, *p* < 0.001), and the interaction between perceived stigma and social support had a significant effect on depressive symptoms (β = −0.188, *p* < 0.001), suggesting that social support moderated the relationship between enacted stigma, perceived stigma, and depressive symptoms (Table 5). The mediation model with moderation is plotted in Figure 2.

The results of the conditional indirect effects showed that the effect of enacted stigma on depressive symptoms increased from 0.740 to 1.577 at changes in social support from low levels (one SD below average) to high levels (one SD above average.) The indirect effect of enacted stigma mediated by perceived stigma on depression decreased from 0.573 to 0.063. Where the indirect effect of enacted stigma on depressive symptoms through perceived stigma was not significant at high levels of social support (b = 0.063, 95% CI = −0.101 to 0.227).

To better explain the mediated model with moderation, simple slope plots were drawn for social support, enacted stigma, perceived stigma, and depressive symptoms by dividing them into high and low subgroups, respectively, with mean ±1 standard deviation. The moderating effect of social support on the relationship between enacted stigma and depression is shown in Figure 3a; the positive effect of enacted stigma on depressive symptoms increased with increasing levels of social support, and the moderating variable reinforced the effect of enacted stigma on depression. The moderating effect of social support on the relationship between perceived stigma and depression is shown in Figure 3b, where the positive effect of perceived stigma on depressive symptoms decreases with increasing levels of social support, and the moderating variable weakens the effect of perceived stigma on depression.

## 4. Discussion

The present study presents a moderated mediation model based on the sexual minority stress model with the aim of exploring the relationship between MSM stigma and depression. The results of the mediated effects model showed that enacted stigma and perceived stigma had a significant positive effect (β = 0.503, *p* < 0.001) (β = 0.204, *p* < 0.001) on depressive symptoms, which is consistent with previous findings [38,39]; stigma is a significant predictor and determinant of depression.

This is an important theoretical model in the field of psychological research that typifies “stress—health,” and we can use sexual minority stress theory to understand the mechanisms by which stigma acts on depression. Meyer views MSM stigma as a form of mental stressor that arises when societal norms conflict with minority cultures and minority interests [40], creating a stressor that, in turn, stimulates the senses and undermines mental health [41].

In addition, this continuous stress process was further confirmed in the present study, where enacted stigma not only affects depression directly but also indirectly through perceived stigma (β = 0.315, 95% CI = 0.221 to 0.421). Societal discrimination and exposure to violence not only lead to the perpetuation of stigma but also to the internalization of negative attitudes and the creation of perceived stigma, the cross-stimulation of which leads to a higher prevalence of depression among men who have sex with men [42].

As a result, it is important to pay more attention to men who have sex with men who are victims of violence and injustice, to educate society about sexual minorities and to promote positive values, and to reduce the stigma that is perpetrated and expected to be perpetrated, which will help to reduce the prevalence of depression among men who have sex with men and to improve their quality of life. In earlier tests, we found that resilience was not significant as a moderator, but as a mediator; it mediated the effect of enacted stigma on depression (β = −0.082, 95% CI = −0.118 to −0.051), and hypothesis (1) was tested. This suggests that resilience is a protective factor for depression, and several studies have found similar results [25,43]. Resilience can be an effective target for intervention and can help reduce the direct and indirect effects of stigma on depressive symptoms.

According to the sexual minority stress model, social support moderated two main pathways: enacted stigma to depression and perceived stigma to depression. The mediation model with moderation resulted in a significant main effect (β = 0.501, *p* < 0.001) (β = 0.170, *p* < 0.001) and a significant interaction term (β = 0.194, *p* < 0.001) (β = 0.188, *p* < 0.001), suggesting that social support moderates the relationship between stigma and depression, and hypothesis (2) was tested.

During the role of perceived stigma on depression, social support weakened the relationship, i.e., as social support increased, the predictive effect of perceived stigma on depression diminished. At high levels of social support, the relationship was no longer significant. MSM who receive high levels of social support are less likely to be depressed and less likely to have expectations that stigma will be perpetrated, which is consistent with previous research findings [31,34,44].

Surprisingly, social support reinforced the positive effect of stigma on depression during the enacted stigma on depression, and the moderating effect was significant at both levels. A plausible explanation for this result is that men who have sex with men receive help and support from friends and family, but when they are in a social environment, they can still be subjected to discrimination and even violence from other people, and these two things are not in conflict. This contradiction between the big external environment (objective stigma) and the small environment around them (positive treatment from family and friends) creates a sense of guilt within them, which leads to an unhealthy psychological condition. There is a need to develop individual and social-level interventions for MSM to leverage social support to reduce the impact of stigma on depression.

These findings have implications for future policy and intervention program development. First, the most immediate and effective approach is to reduce the objective discrimination against MSM in society. Second, we need to consider psychological interventions for MSM at both the individual and societal levels. It is important to guide MSM to develop a positive identity to enhance psychological resilience and to promote care and support from peers and family members. In addition, our study found that social support reinforced the effect of enacted stigma on depression, suggesting that social support is not necessarily a protective factor for depression in some situations. This is a surprising finding and differs from previous findings. In this regard, we suggest that measures need to be taken to balance the internal and external environments of MSM, otherwise support from family and friends may lead to a sense of guilt. Finally, by applying sexual minority stress theory to a sample of Chinese MSM and linking sexual minority stress to depression among Chinese MSM, this study demonstrates that minority stress is an important determinant of depression among Chinese MSM, and to some extent provides insights for improving the mental health of MSM in other parts of the world.

## 5. Limitations

Several limitations should be considered in this study. First, the cross-sectional design of this study does not allow for causal inferences between variables. In addition, all measurements were completed as self-reports. Second, this study utilized a non-probability sampling method, which may be subject to some bias. Finally, the present study’s explanation of the moderating role of social support is, to some extent, based on extrapolation of the results. For example, the positive effect of social support reinforcing the implementation of stigma on depressive symptoms may be due to the conflict inherent in MSM, which needs to be explored more deeply in terms of psychological mechanisms. Additionally, depression among MSM can be influenced by their intrinsic sexual identity, with some studies suggesting that bisexual men are more likely to experience mental health problems than homosexual men, so we need further research to differentiate between gender identities. The average age of MSM recruited in this study was 26.34 years old, which indicates that MSM are showing a lower age in China. Most of the older MSM maintain a stable life status, and they are more reluctant to reveal their sexual identity. The low age of the participants led to the bias of the results of this study, but they are still representative to a certain extent. Finally, there are limitations in extrapolating the results of this study to other countries.

## 6. Implications for Future Research

First, future studies should clarify participants’ sexual orientation and identity when recruiting them. Second, longitudinal studies are needed to establish the causal relationship between stigma and depression, providing an important basis for developing inter ventions. Third, a mixed qualitative and quantitative research methodology to explore the mechanism of stigma on depression from psychological and sociological perspectives will facilitate a more comprehensive understanding of MSM.

## 7. Conclusions

Enacted stigma and perceived stigma can directly influence depressive symptoms and resilience, and perceived stigma partially mediates the relationship between enacted stigma and depressive symptoms. Social support moderated the relationship between enacted stigma, perceived stigma, and depression. Increasing resilience and social support while reducing stigma among MSM may be a measure to alleviate depression among MSM.

## Figures and Tables

**Figure 1 healthcare-11-02849-f001:**
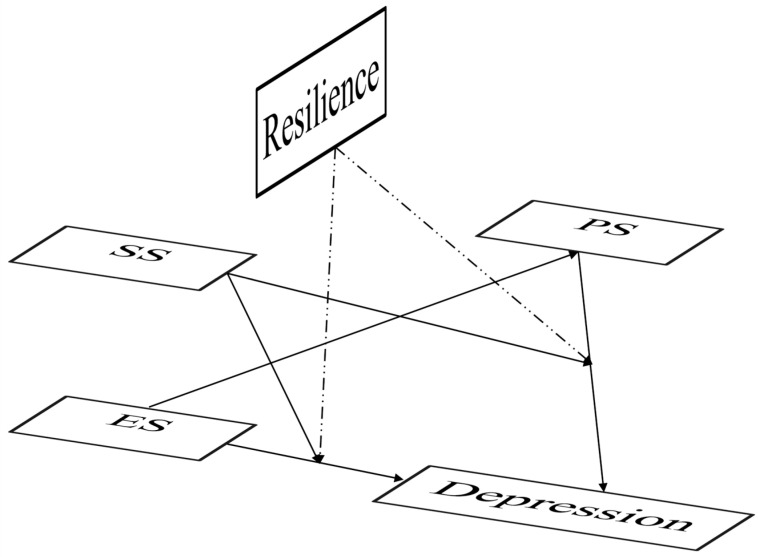
Initial hypothesis model. Note: ES, enacted stigma; PS, perceived stigma; SS, social support. Dotted lines indicate insignificant paths and solid lines indicate significant paths.

**Figure 2 healthcare-11-02849-f002:**
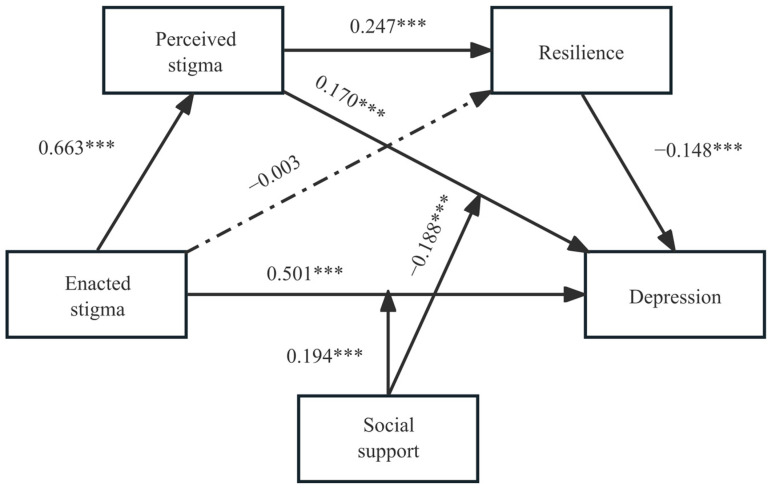
Moderated mediation model. The values shown the standardized coefficients. *** *p* < 0.001. Dotted lines indicate insignificant paths and solid lines indicate significant paths.

**Figure 3 healthcare-11-02849-f003:**
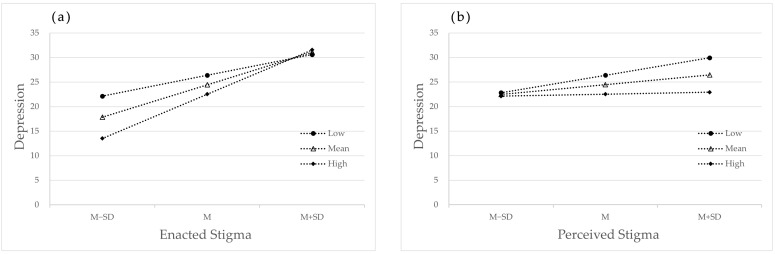
Moderating effect of social support on the stigma and depression. (**a**) Moderating effect of social support on the enacted stigma and depression; (**b**) Moderating effect of social support on the perceived stigma and depression.

**Table 1 healthcare-11-02849-t001:** Characteristics of men who have sex with men survey participants (n = 1091).

Variables	Classification	n	%
Age	18–26	680	62.3
	27–35	320	29.3
	>35	91	8.4
Household registration	Urban	757	69.4
	Rural	334	30.6
Education level	Illiteracy and semi-illiteracy	8	0.7
	Grade school	16	1.5
	Junior high school	60	5.5
	Senior high school	179	16.4
	Junior college	291	26.7
	University degree or above	537	49.2
Employment status	Employed	728	66.7
	Retired	28	2.6
	Students	269	24.7
	Unemployed	66	6.0
Marital status	Unmarried	761	69.8
	Married	293	26.8
	Divorced	32	2.9
	Widowed	5	0.5
Monthly disposable income (RMB)	<1000	49	4.5
	1001–3000	234	21.5
	3001–5000	344	31.5
	5001–10,000	346	31.7
	>10,000	118	10.8

Note: RMB, Ren Min Bi.

**Table 2 healthcare-11-02849-t002:** Correlation of variables (n = 1091).

Variables	Mean	S.D.	1	2	3	4	5	6	7	8
1. Age	26.34	6.324	1							
2. Education	5.14	1.051	−0.094 ***	1						
3. Income	3.23	1.045	0.322 ***	0.192 ***	1					
4. Enacted stigma	15.13	5.717	0.024	−0.182 ***	0.063 **	1				
5. Perceived stigma	18.62	6.230	0.035	−0.026	0.029	0.642 ***	1			
6. Resilience	36.38	8.545	0.550	0.082 ***	0.118 ***	0.148 ***	0.247 ***	1		
7. Social support	44.85	10.331	0.09 ***	0.059	0.102 ***	0.021	0.051	0.663 ***	1	
8. Depression	24.00	13.320	−0.017	−0.240 ***	−0.061 **	0.621 ***	0.475 ***	−0.107 ***	−0.210 ***	1

Note: S.D., standard deviation; ** *p* < 0.01; *** *p* < 0.001.

**Table 3 healthcare-11-02849-t003:** Results of confirmatory factor analysis.

Models	X^2^/df	CFI	TLI	RMSEA	SRMR
Five factors (ES, PS, SS, R, D)	3.567	0.928	0.924	0.049	0.043
Four factors (ES + PS, SS, R, D)	4.898	0.890	0.885	0.060	0.051
Four factors (ES, PS, SS + R, D)	5.981	0.860	0.853	0.068	0.064
Three factors (ES + PS, SS + R, D)	7.294	0.822	0.815	0.076	0.069
Two factors (ES + PS + SS + R, D)	13.620	0.643	0.628	0.108	0.206
Single factor (ES + PS + SS + R + D)	20.896	0.437	0.414	0.135	0.233

Note: ES, enacted stigma; PS, perceived stigma; SS, social support; R, resilience; D, depression. 3.4. Mediation effect analysis; X^2^, chi-square; df, degree of freedom; CFI, comparative fit index; TLI, Tucker–Lewis index; RMSEA, root mean square error of approximation; SRMR, standardized root mean square residual.

**Table 4 healthcare-11-02849-t004:** Results of the test for mediating effects of depression (n = 1091).

Outcome Variable	Predictive Variable	R^2^	F	β	b	S.E.	t	LLCL	ULCL
Model 1									
Perceived stigma	Enacted stigma	0.423	199.159 ***	0.663 ***	0.723	0.025	28.144	0.672	0.773
Model 2									
Resilience	Enacted stigma	0.078	18.490 ***	−0.03	−0.005	0.05	−0.088	−0.120	0.109
	Perceived stigma			0.247 ***	0.339	0.052	6.454	0.236	0.443
Model 3									
Depression	Enacted stigma	0.464	156.734 ***	0.503 ***	1.173	0.069	16.846	1.036	1.310
	Perceived stigma			0.204 ***	0.437	0.063	6.854	0.311	0.562
	Resilience			−0.215 ***	−0.336	0.036	−9.323	−0.407	−0.265
Model 4									
Depression	Enacted stigma	0.408	105.339 ***	0.604 ***	1.408	0.055	25.323	1.299	1.513

Note: S.E., standard error; LLCL, ULCL, upper and lower confidence intervals; *** *p* < 0.001; covariates include age, education level, and monthly disposable income.

**Table 5 healthcare-11-02849-t005:** Conditional process analysis of the proposed moderated mediation model (n = 1091).

Outcome Variable	Predictive Variable	R^2^	F	β	b	S.E.	t	LLCL	ULCL
Model 1									
Perceived stigma	Enacted stigma	0.423	199.159 ***	0.663 ***	0.723	0.025	28.177	0.675	0.776
Model 2									
Resilience	Enacted stigma	0.078	18.490 ***	−0.003	−0.005	0.059	−0.080	−0.120	0.111
	Perceived stigma			0.247 ***	0.339	0.052	6.399	0.234	0.442
Model 3									
Depression	Enacted stigma	0.495	117.537 ***	0.501 ***	1.168	0.069	17.016	1.010	1.279
	Perceived stigma			0.170 ***	0.364	0.630	5.796	0.202	0.459
	Resilience			−0.148 ***	−0.230	0.048	−4.771	−0.308	−0.118
	Social support			−0.124 ***	−0.160	0.038	−4.204	−0.266	−0.110
	Enacted stigma × Social support			0.194 ***	0.039	0.006	6.194	0.027	0.052
	Perceived stigma × Social support			−0.188 ***	−0.030	0.005	−4.582	−0.035	−0.014

Note: *** *p* < 0.001; covariates include age, education level, and monthly disposable income.

## Data Availability

The data presented in this study are available on request from the corresponding authors. The data are not publicly available as they contain sensitive personal behaviors.

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
