# Peer review of "A Study of the Relationship between Men Who Have Sex with Men Stigma and Depression: A Moderated Mediation Model"

_healthcare, 2023, doi:10.3390/healthcare11212849_

Round 1

Reviewer 1 Report

Comments and Suggestions for Authors

This is a great study and well written manuscript what should be published.

From my point of view there is one small but important point to be addressed.  Authors start introduction with a correct sentence "Globally, men who have sex with men (MSM) have significantly poorer mental  health compared to the general population, with a particularly high prevalence of depressive symptoms.".  Of cours, global prevalence is an overage value what strongly depends on the situation in countries with many inhabitants. Such countries are China, India, Russia and some african countires additionally to US.  We know that the situation of men having sex with men very different in democratic versus non-democratic countries. In Russia the situation is catastrophic  as men havon sex with men become persecuted. The same in countries like Iran. In Europa , these men have no problems anymore, they can marry, they have good jobs, no problems.  I am not aware of situation in China or some other Asian countires. 

So I think, authors should

1- mention that sitation strongly differ between countries

2- expalain on one sentence the situation in China

3- add limitations that results cannot be probably extrapolated to other countires

Author Response

Dear Reviewer,

On behalf of my co-authors, we thank you very much for giving us an opportunity to revise our manuscript, and we also appreciate the reviewers very much for their positive and constructive comments and suggestions on our manuscript.

We revised the manuscript according to these comments and suggestions. In general, we have tried our best to revise our manuscript and provide the point-by-point responses. 

Comments 1:[mention that sitation strongly differ between countries]

Response 1: [We agree with this comment, which is why we have emphasized it in the introductory section: The prevalence of depression varies somewhat between countries due to a variety of factors.]

Comments 2:[expalain on one sentence the situation in China]

Response 2: [We have added in the second paragraph of the introductory section the situation of MSM in China, including historical development and the current situation.]

Comments 3:[add limitations that results cannot be probably extrapolated to other countires]

Response 3: [We added in the limitations section. Line 346.]

Thank you again for your valuable comments, attached is the revised manuscript with the changes marked in red.

Reviewer 2 Report

Comments and Suggestions for Authors

The work presented is of enormous interest and relevance. It is worth paying attention to those aspects related to stigma, depression in men who have sex with men.

It is worth thinking about these aspects from the perspective of resilience and social support.

Below we present a series of recommendations to improve the work presented:

- The introduction is very complete and facilitates the reader to understand more and better the stress model and its implications. However, the objective of the paper is not clearly identified. 

- It would be advisable for the authors to explain how they checked the criteria for inclusion in the study: How do they know that the participant does not have an intellectual disability in an online questionnaire format?

- Although the category Men who have sex with men has been widely used for decades, it is increasingly out of use. In this regard, it is surprising that the authors do not ask for the identity of the participants. This would provide much information to inform the model they present. 

- A section on possible future lines of work based on the study would be advisable. 

- It is important that the authors review the bibliographic references. There are important errors in some of them. For example, in reference 3, in reference 4,...

Author Response

Dear Reviewer,

On behalf of my co-authors, we thank you very much for giving us an opportunity to revise our manuscript, and we also appreciate the reviewers very much for their positive and constructive comments and suggestions on our manuscript.

We revised the manuscript according to these comments and suggestions. In general, we have tried our best to revise our manuscript and provide the point-by-point responses. 

Comments 1:[The introduction is very complete and facilitates the reader to understand more and better the stress model and its implications. However, the objective of the paper is not clearly identified. ]

Response 1: [We have added to this in the introduction: The Sexual Minority Stress Model has received much validation in Western samples, but there is less research on depression and sexual minority stress among Chinese MSM. This study aims to suggest mental health interventions for MSM in China, and to some extent provide insights for improving the mental health of MSM in other parts of the world. Line 108.]

Comments 2:[It would be advisable for the authors to explain how they checked the criteria for inclusion in the study: How do they know that the participant does not have an intellectual disability in an online questionnaire format?]

Response 2: [In this study, we collaborated with non-governmental organizations, and most MSM were recruited from cohorts that had been maintained for many years and were of relatively consistent quality.  Rigorous inclusion-exclusion checks were performed at the time of their entry into the cohort, and there was no question of intellectual disability.  All participants were able to complete the survey independently.]

Comments 3:[Although the category Men who have sex with men has been widely used for decades, it is increasingly out of use. In this regard, it is surprising that the authors do not ask for the identity of the participants. This would provide much information to inform the model they present. ]

Response 3: [We agree with this comment, and in the limitations section we add this: Additionally, depression among MSM can be influenced by their intrinsic sexual identity, with some studies suggesting that bisexual men are more likely to experience mental health problems than homosexual men, so we need further research to differentiate between gender identities. Line 338. In addition, we have added to this in future studies that could be conducted. Line 349. Finally, in our other recent study, we included sexual minority identity as a latent variable in a structural equation model.]

Comments 4:[A section on possible future lines of work based on the study would be advisable.]

Response 4: [We add a section called Implications for future research to discuss directions of work that could be considered in the future, including longitudinal research as well as qualitative research. Line 348]

Comments 5:[It is important that the authors review the bibliographic references. There are important errors in some of them. For example, in reference 3, in reference 4,...]

Response 5: [We rechecked the references and revised them to ensure that they were closely related to the study.]

Thank you again for your valuable comments, attached is the revised manuscript with the changes marked in red.

Reviewer 3 Report

Comments and Suggestions for Authors

Thank you for the opportunity to read this highly interesting article. The topic of the study is very relevant, text is a coherent whole, and authors have made careful and important job with this sensitive area of homosexuality. The used variables seem to be good, analysis is well done, and findings are well reported.

I have a few suggestions how the authors could still improve the quality of the text:

1. Introduction should be wider. As a foreign reader, I would need more information about sexual minorities and their rights in Chinese culture. Why is this area very relevant research topic in China? (I am sure that it is very relevant but please, argue why). How are sexual minorities in China nowadays and have there been any improvements/changes in recent history? Give a short historical and cultural report in the introduction.

2. What new does your study bring? You have mentioned many previous studies that support your findings. Please, indicate better what is the big meaning of your study and its findings. 

3. Rethink the limitations of your study. What are the limitations of quantitative research? Could you find something new with qualitative methods or mixed-method study? What are the limitations of the young respondents? Why almost all respondents are very young?

Author Response

Dear Reviewer,

On behalf of my co-authors, we thank you very much for giving us an opportunity to revise our manuscript, and we also appreciate the reviewers very much for their positive and constructive comments and suggestions on our manuscript.

We revised the manuscript according to these comments and suggestions. In general, we have tried our best to revise our manuscript and provide the point-by-point responses. 

Comments 1:[Introduction should be wider. As a foreign reader, I would need more information about sexual minorities and their rights in Chinese culture. Why is this area very relevant research topic in China? (I am sure that it is very relevant but please, argue why). How are sexual minorities in China nowadays and have there been any improvements/changes in recent history? Give a short historical and cultural report in the introduction. ]

Response 1: [We agree with this comment, In the second paragraph of the introductory section, we provide a detailed description of the situation of MSM in China, including the history, culture, and current situation. Line 42.]

Comments 2:[What new does your study bring? You have mentioned many previous studies that support your findings. Please, indicate better what is the big meaning of your study and its findings.]

Response 2: [Our study found some different results: In addition, our study found that social support reinforced the effect of enacted stigma on depression, suggesting that social support is not necessarily a protective factor for depression in some situations. This is a surprising finding and differs from previous findings. In this regard, we suggest that measures need to be taken to balance the internal and external environments of MSM, otherwise support from family and friends may lead to a sense of guilt. Finally, by applying sexual minority stress theory to a sample of Chinese MSM and linking sexual minority stress to depression among Chinese MSM, this study demonstrates that minority stress is an important determinant of depression among Chinese MSM, and to some extent provides insights for improving the mental health of MSM in other parts of the world. These findings have implications for future policy and intervention planning. Line 320.]

Comments 3:[Rethink the limitations of your study. What are the limitations of quantitative research? Could you find something new with qualitative methods or mixed-method study? What are the limitations of the young respondents? Why almost all respondents are very young?]

Response 3: [Quantitative research has the following limitations: the first is that quantitative research emphasizes objectivity and universality at the expense of subjectivity. Secondly, the questionnaires and scales we often use hide many problems. Thirdly quantitative research also runs the risk of unreliable statistics.

We believe that the use of qualitative or mixed research methods can explore the mechanisms of stigma on depression from psychological and sociological perspectives, which is conducive to a more comprehensive understanding of men who have sex with men (MSM) in China. We add to this in the section on Implications for Future Research. Line 348.

Finally, in the section on limitations, we add the following: The average age of MSM recruited in this study was 26.34 years old, which indicates that MSM is showing a lower age in China. Most of the older MSM maintain a stable life status, and they are more reluctant to reveal their sexual identity. The low age of the participants led to the bias of the results of this study, but they are still representative to a certain extent. Finally, there are limitations in extrapolating the results of this study to other countries. Line 342. In all of our team's other studies, MSM showed an overall younger age profile, which we believe indicates that MSM in China is showing a younger age profile.  This makes them representative, but considering that older MSM may be reluctant to reveal their sexual identity due to various factors, too many young participants may also lead to some bias in the results of this study.  We have added this to the limitations.]

Thank you again for your valuable comments, attached is the revised manuscript with the changes marked in red.